# Explanatory Model Based on the Type of Physical Activity, Motivational Climate and Adherence to the Mediterranean Diet of Anxiety among Physical Education Trainee Teachers

Eduardo Melguizo-Ibáñez , Gabriel González-Valero , Pilar Puertas-Molero *, Félix Zurita-Ortega , José Luis Ubago-Jiménez and José Manuel Alonso-Vargas

Department of Didactics of Musical, Plastic and Corporal Expression, University of Granada, 18071 Granada, Spain
* Correspondence: pilarpuertas@correo.ugr.es

**Abstract:** It has now been shown that an active and healthy lifestyle among university students helps to channel disruptive states arising from the academic environment. The objectives of this research are to describe the levels of anxiety, adherence to the Mediterranean diet, and motivational climate as a function of the type of physical activity, and to establish the relationship between adherence to the Mediterranean diet, motivational climate, and anxiety in university students. This objective is broken down into: (a) developing an explanatory model of the motivational climate towards sport and adherence to the Mediterranean diet on anxiety, and (b) contrasting the structural model by means of a multi-group analysis as a function of the type of physical activity. A descriptive, comparative, cross-sectional, and non-experimental study was proposed in a sample of 569 trainee teachers (M = 25.09; SD = 6.22). A sociodemographic questionnaire, the Beck Anxiety Inventory (BAI), the Perceived Motivational Climate in Sport Questionnaire (PMCSQ-2), the PREDIMED Questionnaire, and the International Physical Activity Questionnaire (IPAQ-SF) were used for data collection. The data show that subjects with higher weekly physical activity time show lower levels of anxiety, better adherence to the Mediterranean diet, and demonstrate task-oriented sport motivation.

**Keywords:** healthy lifestyle; disruptive states; active lifestyle

## 1. Introduction

From a psychological and physical perspective, adolescence is a very sensitive phase of human development, where changes begin to take place [1]. This stage begins during puberty and concludes with the acquisition of biopsychosocial maturity, occurring between 10 and 19 years of age [2,3]. With regard to psychological development, adolescents have not yet shaped their personality and are easily influenced by the external context around them [4]. In view of the above, the area of physical education should promote active and healthy lifestyles that last throughout adulthood and help to improve people's health [5].

It has been observed that during adolescence there is a decrease in weekly physical activity times [6], as young people opt for more sedentary activities [7]. Physical activity can be defined as any bodily movement carried out by any type of muscular contraction that results in substantial energy expenditure [8]. A low level of physical activity has a negative impact on young women's health [9], as an active lifestyle has been shown to produce psychological and physical benefits [10]. In this case, benefits such as waist circumference reduction [11], prevention of type 2 diabetes [11], reduction of blood pressure [12], as well as prevention of stroke and cerebral thrombosis [13] have been shown. The physical education classroom has been shown to play a key role in ensuring that physical activity lasts throughout the different stages of development [14], as depending on how the teacher guides the physical education classes, he or she can develop motivation that helps to foster a positive attitude towards such behaviour [15].

Within the physical-sports field, one of the most studied factors is motivation. This concept can be defined as a mechanism that controls the direction and intensity of efforts in order to explain the different human behaviours [16]. One of the most studied theories to explain human behaviour in the performance of a given task is the achievement goal theory [17], where the term motivational climate is proposed [18]. This concept is defined as implicit or explicit signals perceived in the environment through which the keys to success and failure towards a certain task are defined [18]. Within the physical-sports context, when the practice of physical activity is oriented towards mastery, intrinsic values (task climate) acquire greater importance; however, when the activity is emphasised towards extrinsic values, competition is encouraged (ego climate) [19]. In this case, when the practice of physical activity is developed from intrinsic motivation, this tends to be reproduced during the different stages of human development [20]. Likewise, it has been found that when sport practice is developed from extrinsic motivation and the proposed objectives are not achieved, there is an increase in the levels of frustration and anxiety [20], which can lead to the abandonment of the activity due to the feeling of incompetence [20].

Anxiety is another construct that has been studied by psychology over time. It is defined as a state of worry that is difficult to control, where symptoms such as difficulty concentrating, muscle tension, and irritability are associated [21]. Numerous studies indicate that obesity coexists with pathologies such as anxiety [22–24]. In this case, it has been observed that continuous subjection to a state such as anxiety can lead to a response known as over-emotional eating, which consists of an increased intake of unhealthy foods [25], and this behaviour is associated with low emotional control [26].

Based on the above, the Mediterranean diet has been shown to be a healthy dietary pattern that has a positive physical and emotional impact [3]. This dietary pattern is characterised by a decrease in the consumption of saturated fats, with a greater presence of monounsaturated fatty acids such as olive oil [27]. The foods that make up this dietary pattern are those characteristics of the Mediterranean area, such as vegetables, fruit, cereals, and legumes [27]. It has been shown that during adolescence there is a shift towards a healthy pattern as young people begin to have greater control over their diet [28]; however, it has been observed that poor nutrition and health education has a negative impact on the health of young people, as a greater number of adolescents are obese and suffer from cardiovascular diseases [29].

In view of the above, the present study aims to describe the levels of anxiety, adherence to the Mediterranean diet, and motivational climate as a function of the type of physical activity, and to establish the relationship between adherence to the Mediterranean diet, motivational climate, and anxiety in university students. This objective is broken down into: (a) developing an explanatory model of the motivational climate towards sport and adherence to the Mediterranean diet on anxiety, and (b) contrasting the structural model through a multi-group analysis as a function of the type of physical activity.

Finally, the research hypotheses that we propose are stated below:

**H1.** *Participants who show a low level of weekly physical activity will show higher levels of anxiety than participants who show a moderate or high level of weekly physical activity.*

**H2.** *Young people who show a low level of weekly physical activity will show worse adherence to the Mediterranean diet than participants who show a moderate or high level of weekly physical activity.*

**H3.** *Subjects who show a high level of weekly physical activity will have a negative relationship between anxiety and adherence to the Mediterranean diet.*

**H4.** *There will be a positive relationship between the two motivational climates and adherence to the Mediterranean diet.*

**H5.** *Task climate will be negatively associated with anxiety.*

## 2. Materials and Methods

### 2.1. Design and Participants

A quantitative, comparative, and non-experimental (ex post facto) study was carried out following a cross-sectional design. The study sample consisted of a total of 569 trainee teachers (M = 25.09; SD = 6.22) belonging to the Faculty of Education Sciences of the University of Granada. Regarding gender distribution, 141 belong to the male sex and 428 to the female sex. Regarding the sampling error, for a maximum confidence error of 95%, an error of 3.15% was reached.

### 2.2. Instruments

**Sociodemographic Questionnaire:** Aimed at collecting sociodemographic variables such as the sex of the participants (male or female) and age.

**Beck Anxiety Inventory (BAI):** This instrument was developed by Beck et al. [30], but the Spanish version adapted by Sanz and Navarro [31] has been used. The questionnaire consists of a total of 21 items (I feel unsteady), which are evaluated on a Likert scale (0 = not at all to 3 = very much). In terms of the reliability of this instrument, a score of $\alpha = 0.969$ was obtained.

**Perceived Motivational Climate in Sport Questionnaire (PMCSQ-2):** This questionnaire was developed by Newton et al. [32]; however, we used the version adapted to the Spanish population by González-Cutre et al. [33]. This instrument is made up of a total of 33 items (The coach believes that all of us are crucial to the success of the team/squad) that are evaluated on a Likert scale (1 = strongly disagree and 5 = strongly agree). The questionnaire assesses motivation from the perspective of two motivational climates. One is the task climate, which is composed of three sub-variables: effort enhancement (EI), cooperative learning (CL), and important role (IR). The second motivational climate is the ego climate which is made up of three sub-dimensions: unequal recognition (UR), punishment for mistakes (PM), and rivalry between members (MR). In terms of reliability analysis, a value of $\alpha = 0.958$ was obtained for the task climate, while a score of $\alpha = 0.921$ was obtained for the ego climate.

**PREDIMED Questionnaire:** This was developed by Schröder et al. [34]; however, the Spanish version adapted by Álvarez-Álvarez et al. [35] was used in this study. This instrument is composed of 14 items (How many servings of butter, margarine, or cream do you consume per day?), where once answered, a final score is obtained that categorises participants' responses into three levels: low adherence ($\leq 7$), medium adherence (8–10), and high adherence ($\leq 10$). In this case, Cronbach's Alpha obtained a value of $\alpha = 0.806$.

**International Physical Activity Questionnaire (IPAQ-SF):** The version adapted to Spanish by Mantilla-Toloza and Gómez-Conesa [36] was used for this study. This instrument collects the time (minutes) and frequency (days) dedicated to activities of different intensities. Depending on the final score obtained, the responses are classified into three levels: low, moderate, and high. Finally, Cronbach's alpha obtained a value of $\alpha = 0.830$.

### 2.3. Procedure and Ethical Aspects

Before starting to collect the data, a systematic review and a bibliographical review were carried out in order to study the problems addressed in greater depth. Then, from the Department of Didactics of Musical, Plastic and Corporal Expression of the University of Granada, we proceeded to create a Google Form with the instruments described above. The research objectives were also included. The virtual medium was used for data collection due to the mobility restrictions imposed by the COVID-19 virus. To ensure that responses were not answered in a random manner, two questionnaires were duplicated, eliminating responses from participants where the answers did not match. In this case, a total of 32 questionnaires were eliminated as they were incorrectly completed.

In addition, convenience sampling was carried out, with the inclusion criterion being that participants had to have or be studying for a degree in primary education with a major

in physical education. Failure to meet the above criterion meant direct exclusion from the study.

With regard to the ethical principles followed by the study, the criteria established in the Helsinki Declaration of 1975 were followed at all times. Furthermore, the present research was approved and supervised by an ethics committee belonging to the University of Granada (2966/CEIH/2022). Finally, the participants gave their written informed consent and were able to withdraw from the research at any time. Participants were assured that the data would be processed anonymously and for scientific purposes only.

*2.4. Data Analysis*

The IBM SPSS 25.0 statistical programme (SPSS, IBM, SPSS Statistics, v.25.0 Chicago, IL, USA) was used for the comparative analysis. The analysis of the normality of the variables was carried out together with the study of the homogeneity of the sample by means of the Kolmogorov–Smirnov test. After checking that the variables followed a normal distribution, the data were analysed by means of the one-factor ANOVA test, determining the statistically significant differences by means of Pearson's chi-squared test. In this case, the significance level was set at $p \leq 0.05$. To study the magnitude of the effect size (ES) difference, Cohen's standardised d-index [37] was used, interpreted as null ($\leq 0.19$), small (0.20–0.49), medium (0.50–0.79), and large ($\geq 0.80$).

To develop the structural equation models, the statistical software IBM SPSS Amos 26.0 (IBM Corp., Armonk, NY, USA) was used. For this research, a model was developed for each of the different categories that make up the practice of physical activity. Each model is composed of two exogenous variables (TC; EC) and eight endogenous variables (CL; EI; IR; PM; UR; MR; MDA; ANX). For the first type of variables, a causal explanation has been carried out on the basis of the associations observed between the indicators and the reliability of measurement, which is why measurement error has been included in the different models (Figure 1). Regarding the direction of the arrows, the unidirectional arrows symbolise the lines of influence between the latent variables, which are interpreted from the regression weights. Regarding the level of significance, two levels were established: one at $p \leq 0.05$ and the other one at $p \leq 0.001$.

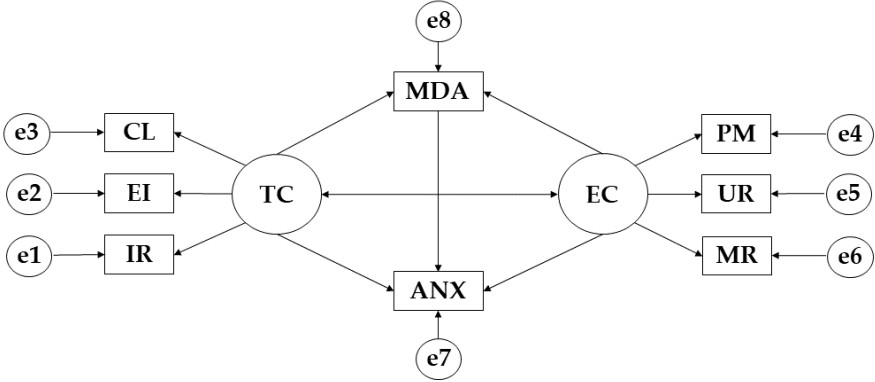

**Figure 1.** Proposed theoretical model. **Note:** EC = Ego climate; TC = Task climate; PM = Punishment for mistakes; UR = Unequal recognition; MR = Member rivalry; CL = Cooperative learning; EI = Effort/improvement; IR = Important role; ANX = Anxiety; MDA = Mediterranean diet adherence.

To evaluate the different models proposed, we proceeded to study the scores on the different fit parameters. Following the criteria established by Bentler [38] and McDonald and Marsh [39], the goodness of fit should be assessed through the chi-square test, where non-significant values are evidence of a good fit. Furthermore, following the recommendations proposed by Kiriazos [40], the comparative fit index (CFI), the goodness-of-fit index

(GFI), and the incremental reliability index (IFI) should be higher than 0.900. Finally, the root mean square approximation (RMSEA) value should be less than 0.100.

## 3. Results

### 3.1. Comparative Analysis

Table 1 shows the comparative analysis of the variables according to the level of physical activity. For MDA, statistically significant differences are observed between participants with a low and moderate level of physical activity. In this case, participants with a moderate level ($0.50 \pm 0.121$) show a higher degree of adherence to this dietary pattern than those with a low level ($0.48 \pm 0.135$). With regard to ANX, statistically significant differences were observed between participants with a low and moderate level of physical activity. It is observed that those with a low level ($0.97 \pm 0.699$) show higher levels of anxiety than those with a medium level of physical activity ($0.77 \pm 0.582$). Likewise, statistically significant differences are also shown between participants who show a low and high level of weekly physical activity, with participants who show a low level ($0.97 \pm 0.699$) reflecting higher levels of anxiety than those who show a high level of physical activity ($0.71 \pm 0.618$). Regarding CL, differences are evident for the groups with low and medium levels of physical activity. In this case, participants with a moderate level ($4.00 \pm 0.876$) show higher scores than those with a low level of physical activity ($3.80 \pm 1.018$). Continuing with E/I, statistically significant differences are observed between participants showing a low and moderate level of physical activity. In this case, participants showing a moderate level ($3.98 \pm 0.670$) show higher scores than those showing a low level ($3.77 \pm 0.844$). Statistically significant differences are also shown between participants showing a low and high level of weekly physical activity, with participants showing a high level ($4.01 \pm 0.665$) showing higher scores than those showing a low level of physical activity ($3.77 \pm 0.844$). Finally, for IR, significant differences are observed between the high and low physical activity groups. Participants with a high level of physical activity ($4.15 \pm 0.745$) scored higher than those with a low level of weekly physical activity ($3.86 \pm 0.926$).

**Table 1.** Comparative study of the sample in terms of weekly physical activity level.

|  |  | N | M | SD | F | *P* | ES | 95%CI |
|---|---|---|---|---|---|---|---|---|
| **MDA** | Low | 128 | 0.48 | 0.135 | | | | |
| | Moderate | 169 | 0.50 | 0.121 | 4.095 | $\leq$0.05 [b] | 0.318 [b] | [0.107; 0.530] [b] |
| | High | 271 | 0.52 | 0.123 | | | | |
| **ANX** | Low | 128 | 0.97 | 0.699 | | | | |
| | Moderate | 170 | 0.77 | 0.582 | 7.688 | $\leq$0.05 [a,b] | 0.332 [a] | [0.101; 0.563] [a] |
| | High | 271 | 0.71 | 0.618 | | | 0.403 [b] | [0.191; 0.615] [b] |
| **CL** | Low | 128 | 3.80 | 1.018 | | | | |
| | Moderate | 170 | 4.00 | 0.876 | 6.509 | $\leq$0.05 [b] | 0.399 [b] | [0.187; 0.611] [b] |
| | High | 271 | 4.14 | 0.762 | | | | |
| **EI** | Low | 128 | 3.77 | 0.844 | | | | |
| | Moderate | 170 | 3.98 | 0.670 | 5.011 | $\leq$0.05 [a,b] | 0.280 [a] | [0.05; 0.511] [a] |
| | High | 271 | 4.01 | 0.665 | | | 0.329 [b] | [0.116; 0.542] [b] |
| **IR** | Low | 128 | 3.86 | 0.926 | | | | |
| | Moderate | 170 | 4.03 | 0.895 | 5.280 | $\leq$ 0.05 [b] | 0.359 [b] | [0.148; 0.571] [b] |
| | High | 271 | 4.15 | 0.745 | | | | |
| **PM** | Low | 128 | 2.32 | 0.819 | | | | |
| | Moderate | 170 | 2.44 | 0.823 | 0.779 | $\geq$0.05 | NP | NP |
| | High | 271 | 2.38 | 0.770 | | | | |
| **UR** | Low | 128 | 2.72 | 1.049 | | | | |
| | Moderate | 170 | 2.72 | 1.020 | 0.487 | $\geq$0.05 | NP | NP |
| | High | 271 | 2.64 | 1.015 | | | | |
| **MR** | Low | 128 | 2.63 | 0.890 | | | | |
| | Moderate | 170 | 2.78 | 0.883 | 1.029 | $\geq$0.05 | NP | NP |
| | High | 271 | 2.73 | 0.935 | | | | |

**Note 1:** [a] Differences between low and moderate; [b] Differences between low and high.
**Note 2:** PM = Punishment for mistakes; UR = Unequal recognition; MR = Member rivalry; CL = Cooperative learning; EI = Effort/improvement; IR = Important role; ANX = Anxiety; MDA = Mediterranean diet adherence.

*3.2. Structural Equations Model Analysis*

　　Looking at the structural equation models, the model developed for participants showing a low level of weekly physical activity shows a good fit for each of the indices. The chi-square test showed a non-significant *p*-value ($X^2$ = 47.631; df = 16; pl = 0.000), but these data cannot be interpreted in an anonymous way due to the size and susceptibility of the sample [41], so other indices were used to study the fit of the model. In this case, the CFI, NFI, IFI, and TLI obtained scores above 0.910. In addition, RMSEA obtained a value of 0.037.

　　Figure 2 and Table 2 show the effects of the variables for participants showing a low level of weekly physical activity. In this case, a positive relationship is observed between MDA and TC ($p \leq 0.05$; r = 0.204). A positive relationship is also shown between adherence MDA and EC (r = 0.029). Regarding TC, a positive relationship is observed with IR (r = 0.938), (EI; $p \leq 0.001$; r = 0.816), and (CL; $p \leq 0.001$; r = 0.848). Continuing with EC, positive relationships are obtained with PM (r = 0.826), unequal recognition UR ($p \leq 0.001$; r = 0.900), and MR ($p \leq 0.001$; r = 0.488). The relationship between ANX and MDA shows a positive relationship ($p \leq 0.05$; r = 0.038). Focusing attention on anxiety ANX and TC, a negative relationship is observed between both variables (r = −0.078). On the contrary, a positive relationship is obtained between ANX and ego climate EC ($p \leq 0.05$; r = 0.183). Finally, a negative relationship is observed between the two motivational climates ($p \leq 0.001$; r = −0.574).

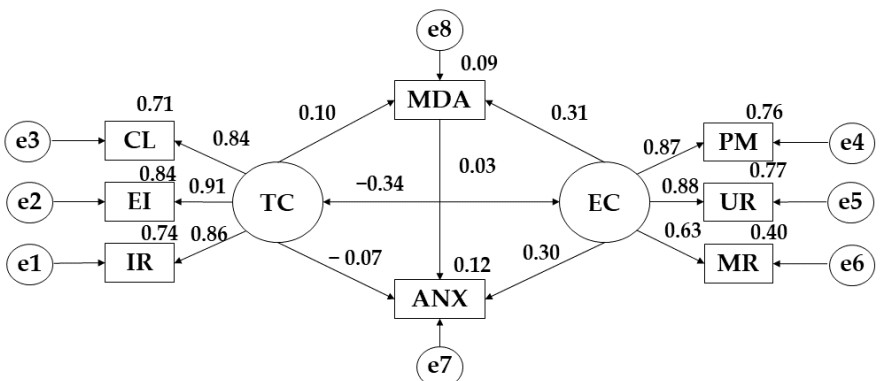

**Figure 2.** Theoretical model proposed for participants showing a low level of weekly PA. **Note:** EC = Ego climate; TC = Task climate; PM = Punishment for mistakes; UR = Unequal recognition; MR = Member rivalry; CL = Cooperative learning; EI = Effort/improvement; IR = Important role; ANX = Anxiety; MDA = Mediterranean diet adherence.

**Table 2.** Structural model for participants showing a low level of weekly PA.

| Associations between Variables | R.W. | | | | S.R.W. |
|---|---|---|---|---|---|
| | Estimations | S.E. | C.R. | *p* | Estimations |
| MDA ← TC | 0.036 | 0.014 | 2.490 | ** | 0.101 |
| MDA ← EC | 0.006 | 0.016 | 0.344 | 0.731 | 0.310 |
| IR ← TC | 1.000 | | | | 0.859 |
| EI ← TC | 0.775 | 0.044 | 17.649 | *** | 0.914 |
| CL ← TC | 0.925 | 0.049 | 18.848 | *** | 0.840 |
| PM ← EC | 1.000 | | | | 0.869 |
| UR ← EC | 1.440 | 0.112 | 12.833 | *** | 0.880 |
| MR ← EC | 0.715 | 0.093 | 7.735 | *** | 0.631 |
| ANX ← MDA | 0.189 | 0.308 | 0.613 | ** | 0.030 |

**Table 2.** *Cont.*

| Associations between Variables | R.W. | | | | S.R.W. |
| --- | --- | --- | --- | --- | --- |
| | Estimations | S.E. | C.R. | *p* | Estimations |
| ANX ← TC | −0.069 | 0.073 | −0.953 | 0.341 | −0.068 |
| ANX ← EC | 0.177 | 0.081 | 2.191 | ** | 0.303 |
| EC ←→ TC | −0.255 | 0.037 | −6.846 | *** | −0.342 |

**Note 1:** EC = Ego climate; TC = Task climate; PM = Punishment for mistakes; UR = Unequal recognition; MR = Member rivalry; CL = Cooperative learning; EI = Effort/improvement; IR = Important role; ANX = Anxiety; MDA = Mediterranean diet adherence. **Note 2:** ** $p \leq 0.05$; *** $p \leq 0.001$.

The model developed for participants showing an average level of weekly physical activity shows a good fit for each of the indices. The chi-square test showed a non-significant *p*-value ($X^2$ = 44.836; df = 16; pl = 0.000). Similarly, the CFI, NFI, IFI, and TLI obtained higher scores of 0.910. In addition, RMSEA showed a value of 0.053.

Figure 3 and Table 3 show the effects of the variables for participants with a low level of weekly physical activity. In this case, a positive relationship is observed between MDA and TC ($p \leq 0.05$; r = 0.128), with exactly the same occurring between MDA and EC (r = 0.054). In terms of TC, positive relationships are obtained with IR (r = 0.911), EI ($p \leq 0.001$; r = 0.816), and CL ($p \leq 0.001$; r = 0.899). Positive relationships are also observed between EC and PM (r = 0.817), UR ($p \leq 0.001$; r = 0.918), and MR ($p \leq 0.001$; r = 0.654). In terms of ANX, a negative relationship is observed with TC ($p \leq 0.05$; r = −0.160). On the other hand, a positive relationship is observed between ANX and EC (r = 0.053) and MDA ($p \leq 0.05$; r = 0.099). Finally, a negative relationship is shown between EC and TC ($p \leq 0.001$; r = −0.541).

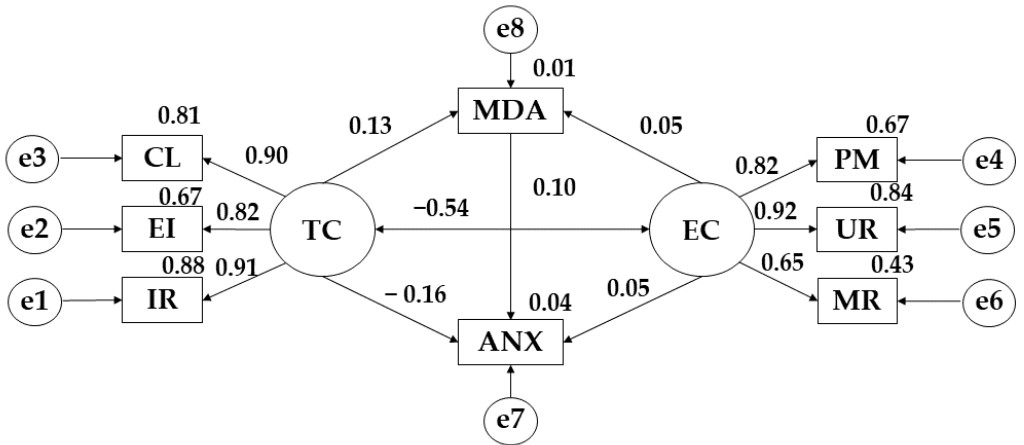

**Figure 3.** Theoretical model proposed for participants showing a medium level of weekly PA. **Note:** EC = Ego climate; TC = Task climate; PM = Punishment for mistakes; UR = Unequal recognition; MR = Member rivalry; CL = Cooperative learning; EI = Effort/improvement; IR = Important role; ANX = Anxiety; MDA = Mediterranean diet adherence.

Looking at the model developed for participants showing a high level of weekly physical activity, it shows a good fit for each of the indices. The chi-square test showed a non-significant *p*-value ($X^2$ = 43.361; df = 16; pl = 0.000), the CFI, NFI, IFI, and TLI scored above 0.890. In addition, the root mean square error of approximation analysis (RMSEA) showed a value of 0.079.

**Table 3.** Structural model for participants showing a medium level of weekly PA.

| Associations between Variables | R.W. | | | | S.R.W. |
|---|---|---|---|---|---|
| | Estimations | S.E. | C.R. | *p* | Estimations |
| MDA ← TC | 0.019 | 0.015 | 1.293 | 0.196 | 0.128 |
| MDA ← EC | 0.010 | 0.018 | 0.537 | 0.591 | 0.054 |
| IR ← TC | 1.000 | | | | 0.911 |
| EI ← TC | 0.672 | 0.048 | 14.057 | *** | 0.816 |
| CL ← TC | 0.970 | 0.059 | 16.406 | *** | 0.899 |
| PM ← EC | 1.000 | | | | 0.817 |
| UR ← EC | 1.402 | 0.122 | 11.487 | *** | 0.918 |
| MR ← EC | 0.858 | 0.097 | 8.873 | *** | 0.654 |
| ANX ← TC | −0.114 | 0.070 | −1.636 | ** | −0.160 |
| ANX ← EC | 0.046 | 0.085 | 0.541 | 0.589 | 0.053 |
| ANX ← MDA | 0.475 | 0.365 | 1.301 | ** | 0.099 |
| EC ←→ TC | −0.296 | 0.056 | −5.303 | *** | −0.541 |

**Note 1:** EC = Ego climate; TC = Task climate; PM = Punishment for mistakes; UR = Unequal recognition; MR = Member rivalry; CL = Cooperative learning; EI = Effort/improvement; IR = Important role; ANX = Anxiety; MDA = Mediterranean diet adherence **Note 2:** ** $p \leq 0.05$; *** $p \leq 0.001$.

Figure 4 and Table 4 show the effects of the variables for participants showing a low level of weekly physical activity. A positive relationship is observed between MDA and TC ($p \leq 0.05$; r = 0.204), with exactly the same occurring between MDA and EC (r = 0.029). Regarding TC, a positive relationship is obtained with IR (r = 0.938), EI ($p \leq 0.001$; r = 0.816), and CL ($p \leq 0.001$; r = 0.848). Continuing with EC, it shows positive relationships with PM (r = 0.826), UR ($p \leq 0.001$; r = 0.900), and MR ($p \leq 0.001$; r = 0.488). With regard to ANX, this disruptive state shows positive relationships with MDA ($p \leq 0.05$; r = 0.038) and EC ($p \leq 0.05$; r = 0.183), while it shows a negative relationship with TC (r = −0.078). Finally, a negative relationship is shown between both motivational climates ($p \leq 0.001$; r = −0.574).

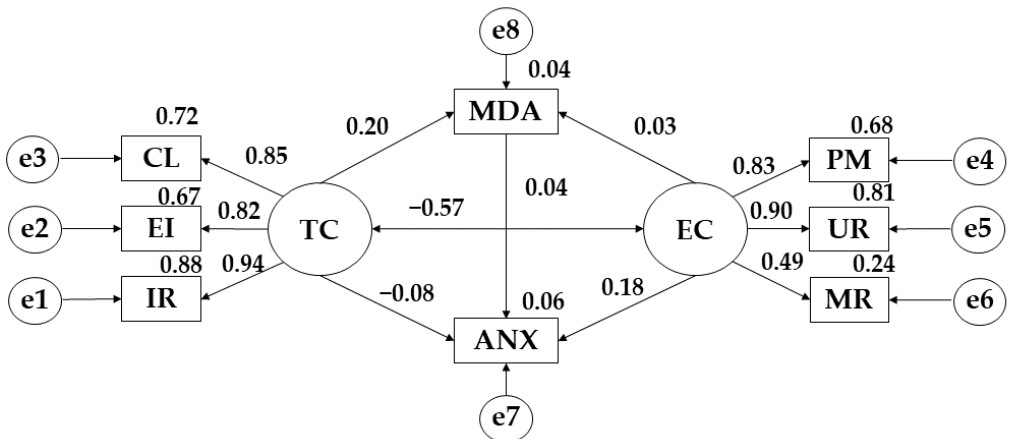

**Figure 4.** Theoretical model proposed for participants showing a high level of weekly PA. **Note:** EC = Ego climate; TC = Task climate; PM = Punishment for mistakes; UR = Unequal recognition; MR = Member rivalry; CL = Cooperative learning; EI = Effort/improvement; IR = Important role; ANX = Anxiety; MDA = Mediterranean diet adherence.

**Table 4.** Structural model for participants showing a high level of weekly PA.

| Associations between Variables | R.W. | | | | S.R.W. |
|---|---|---|---|---|---|
| | Estimations | S.E. | C.R. | *p* | Estimations |
| MDA ←TC | 0.036 | 0.014 | 2.490 | ** | 0.204 |
| MDA ←EC | 0.006 | 0.016 | 0.344 | 0.731 | 0.029 |
| IR ←TC | 1.000 | | | | 0.938 |
| EI ←TC | 0.775 | 0.044 | 17.649 | *** | 0.816 |
| CL ←TC | 0.925 | 0.049 | 18.848 | *** | 0.848 |
| PM ←EC | 1.000 | | | | 0.826 |
| UR ←EC | 1.440 | 0.112 | 12.833 | *** | 0.900 |
| MR ← EC | 0.715 | 0.093 | 7.735 | *** | 0.488 |
| ANX ←MDA | 0.189 | 0.308 | 0.613 | 0.540 | 0.038 |
| ANX ←TC | −0.069 | 0.073 | −0.953 | 0.341 | −0.078 |
| ANX ←EC | 0.177 | 0.081 | 2.191 | ** | 0.183 |
| EC ←→TC | −0.255 | 0.037 | −6.846 | *** | −0.574 |

**Note 1:** EC = Ego climate; TC = Task climate; PM = Punishment for mistakes; UR = Unequal recognition; MR = Member rivalry; CL = Cooperative learning; EI = Effort/improvement; IR = Important role; ANX = Anxiety; MDA = Mediterranean diet adherence. **Note 2:** ** $p \leq 0.05$; *** $p \leq 0.001$.

## 4. Discussion

The present research shows the relationships between adherence to the Mediterranean diet, motivational climate, and anxiety as a function of the type of weekly physical activity. For this reason, the present discussion aims to compare the results obtained with those of other studies already carried out.

In this case, it is observed that participants with a high level of weekly physical activity show a higher degree of adherence to the Mediterranean diet than those with a moderate or low level. Similar results were found by Bonofiglio [42], stating that when a high level of physical activity is carried out, there is a tendency to improve the diet, this being due to the practice of sport at a competitive level. In contrast, very distant results were found by Malakou et al. [43], with Ali et al. [44] stating the need for more nutritional training in the educational environment.

With regard to anxiety, it is observed that participants who show a low level of physical activity show higher levels of this disruptive state than those who show a moderate or high level of weekly physical activity. Very similar results have been found by Melguizo-Ibáñez et al. [9], with Ubago-Jiménez et al. [45] affirming that the regular practice of physical-sports exercise helps to channel disruptive states due to the segregation of neurotransmitters such as serotonin and dopamine. On the contrary, Paolucci et al. [46] state that not all types of physical activity help to channel anxiety. In this case, when continuous training of moderate intensity occurs, there is a decrease in anxiety levels, whereas for high-intensity interval training, an increase in anxiety is observed [46].

Focusing attention on the three variables that make up the task climate, it is observed that participants who show a high level of weekly physical activity show better scores than those who report practising a moderate or low level. In view of these findings, Castro-Sánchez et al. [47] establish that the figure of the coach or teacher plays a key role in encouraging physical activity through intrinsic values. In addition, the study by González-Valero et al. [48] affirms that when the practice of physical activity originates from intrinsic motivation, values such as fun and the enjoyment of physical activity are prioritised. Likewise, it has also been shown that physical activity practised in a group helps to promote values such as companionship [48].

Regarding the variables that make up the ego-climate, it is observed that participants who show a moderate level show greater recognition than subjects who show a

low or high level of weekly physical activity. In this case, similar results were found by Castro-Sánchez et al. [49], stating that when sport practice originates from extrinsic motivation, there is a decrease in physical exercise times. Likewise, when the practice of physical exercise is guided by extrinsic values, factors such as egocentrism among the members of the group to demonstrate a greater degree of competence compared to the rest acquire a greater degree of relevance [50].

Continuing with the structural equation models developed, it is observed that participants who show a high level of weekly physical activity show a higher score than those who show an intermediate or low level. In view of these findings, Cobo-Cuenca et al. [51] affirm that when physical-sports practice originates from intrinsic motivation, care of body image is encouraged, with the Mediterranean diet being a favourable dietary pattern for the care of this body aspect. Very distant results were obtained by Melguizo-Ibáñez et al. [28], who stated that young people spend very little time on the quality of food cooking, prioritising the food offered by fast food apps.

For the ego-climate, a greater association is observed for participants who show a low level of weekly physical activity. Very distant results were found by González-Valero et al. [52], stating that when physical-sports practice is oriented towards competition, greater care is taken in the dietary pattern followed in order to achieve greater performance.

It is also observed that participants who show a moderate level of physical activity show a better association between anxiety and task climate. A study by Ubago-Jiménez et al. [45] concludes that physical-sports practice developed from intrinsic motivation helps to channel anxiety levels, due to the secretion of neurotransmitters, which help to alleviate the effects generated by this disruptive state. On the contrary, it is observed that young people who show a low level of physical activity show a greater association between ego-climate and anxiety. This result coincides with the findings of Orumiyehei et al. [53], who state that high-intensity physical exercise of short duration generates a higher level of anxiety in participants.

Regarding the association between anxiety and adherence to the Mediterranean diet, better scores are observed for participants who show a moderate level of weekly physical exercise. Very different results have been found by Trigueros et al. [54], establishing that a high level of physical activity together with optimal adherence to the Mediterranean diet helps to channel and prevent increased levels of anxiety. Likewise, a study by López-Olivares et al. [55] states that the type of physical activity can promote anxiety levels, with higher intensity physical activity promoting higher levels of this disruptive state.

## 5. Limitations and Future Perspectives

This research has several limitations, the most significant of which are highlighted below. The first of these lies in the typology of the study, as it is a cross-sectional study which only allows us to establish the cause-effect relationships of the variables at that point in time. Moreover, the participants belong to a very specific geographical area, so generalisations cannot be made to a wider area of the national or regional geography. Likewise, the sample is homogeneous in nature, as more than half of the two thirds of the total sample are female.

With regard to the strengths of this study, the data are totally reliable as fully validated instruments adapted to the study population have been used.

Finally, with regard to future perspectives, a longitudinal study is being developed to focus on active and healthy lifestyles in the emotional sphere.

## 6. Conclusions

The present study shows the relationships between motivational climate and adherence to the Mediterranean diet on anxiety as a function of the level of weekly physical activity in a sample of university students.

The comparative analysis shows that participants with a high level of weekly physical activity show lower levels of anxiety. It is also observed that subjects with a high level

of physical activity show a better adherence to the Mediterranean diet. Regarding the motivational climate, participants who show a high level of weekly physical-sports activity show higher scores in the three variables that make up the task climate (cooperative learning, effort/improvement, and important role), while for the ego climate, participants who show a moderate level show higher results.

In terms of the structural equation models, a positive relationship is observed between the two motivational climates and adherence to the Mediterranean diet. Likewise, a positive relationship is also observed between anxiety and adherence to the Mediterranean diet and this disruptive state and ego-climate. In contrast, negative relationships are obtained between anxiety and task climate and both motivational climates.

It is observed that participants with a high level of physical activity have a better relationship between adherence to the Mediterranean diet and task climate. Likewise, subjects with a medium level of physical activity show stronger relationships between anxiety and task climate and Mediterranean diet and anxiety. Finally, participants with a low level of weekly physical activity show better associations between ego-climate and adherence to the Mediterranean diet and between motivational climate and anxiety.

**Author Contributions:** Conceptualization, E.M.-I., J.M.A.-V. and J.L.U.-J.; methodology, P.P.-M.; software, G.G.-V. and J.L.U.-J.; validation, E.M.-I., P.P.-M. and J.M.A.-V.; formal analysis, G.G.-V.; investigation, J.M.A.-V.; resources, E.M.-I; data curation, P.P.-M. and F.Z.-O.; writing—original draft preparation, G.G.-V.; writing—review and editing, J.M.A.-V.; visualization, E.M.-I; supervision, G.G.-V. and F.Z.-O. All authors have read and agreed to the published version of the manuscript.

**Funding:** This research received no external funding.

**Institutional Review Board Statement:** The study was conducted in accordance with the Declaration of Helsinki and approved by the Research Ethics Committee of the University of Granada (2966/CEIH/2022); the approval date is 27 September 2022.

**Informed Consent Statement:** Informed consent was obtained from all subjects involved in the study.

**Data Availability Statement:** The data used to support the findings of the current study are available from the corresponding author upon request.

**Conflicts of Interest:** The authors declare no conflict of interest.

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
