# Peer review of "Explanatory Model Based on the Type of Physical Activity, Motivational Climate and Adherence to the Mediterranean Diet of Anxiety among Physical Education Trainee Teachers"

_applsci, doi:10.3390/app122413016_

Round 1

Reviewer 1 Report

The article title and abstract are appropriate. The purpose of the article and its significance is stated clearly. The study methods are sound and appropriate. The writing is clear and concise. The conclusions or summary are accurate and supported by the content.

The article is of interest to members of the education research community.

I recommend Research Article entitled: ”Explanatory model based on the type of physical activity, motivational climate and adherence to the Mediterranean diet of anxiety among physical education trainee teachers” for publication.

Author Response

REVIEWER 1

Comment 1

The article title and abstract are appropriate. The purpose of the article and its significance is stated clearly. The study methods are sound and appropriate. The writing is clear and concise. The conclusions or summary are accurate and supported by the content.

The article is of interest to members of the education research community.

I recommend Research Article entitled: ”Explanatory model based on the type of physical activity, motivational climate and adherence to the Mediterranean diet of anxiety among physical education trainee teachers” for publication.

Response 1

Thank you very much for your comment. In this case the research has been carried out with the highest possible scientific rigour.

Regarding your suggestion, the authors agree with your recommendation and the title has been modified.

Reviewer 2 Report

Some suggestions for improvement are indicated in the attached document.

Author Response

REVIEWER 2

Comment 1

Line 111: “…a total of 21 items…” replace: “…a total of 21 items (e.g., “indicate an example item” Line 111: “0 = not at all and 3 = very much” replace by: 0 =not at all to 3 = very much. Line 112: α=0.969 replace: α = .969 Make the same changes in the description of the remaining instruments.

Response 1

Thank you very much for your comment. The changes have been implemented

Comment 2

Line 183: I suggest that the authors remove the errors from the figure. Line 183.

Response 2

Thank you very much for your suggestion. In this case the authors consider that this comment cannot be implemented as the error needs to be present to help understand the models. Also, due to the characteristics and recommendations proposed by Tenembaun & Eklund (2007), these need to be present.

Comment 3

Note: Ego climate (EC); Task climate (TC); Punishment for 183 mistakes (PM); Unequal recognition (UR); Member rivalry (MR); Cooperative learning (CL); Ef-184 fort/improvement (EI); Important role (IR); Anxiety (ANX); Mediterranean Diet Adherence (MDA). Replace by: EC=Ego climate; TC=Task climate; PM=Punishment for 183 mistakes; UR=Unequal recognition; MR=Member rivalry; CL=Cooperative learning; EI=Ef-184 fort/improvement; IR=Important role; ANX=Anxiety; MDA=Mediterranean Diet Adherence. Make the same change in the remaining notes of figures and tables.

Response 3

Thank you very much for your suggestion. The changes have been implemented successfully.

Comment 4

Line 192: Eliminate Mediterranean diet (MDA)

Response 4

Thank you very much for your suggestion. The changes have been implemented successfully.

Comment 5

Line 196, 202…: authors must change the acronyms, see the acronyms rule. The first time they are indicated and they always indicate only the acronym.

Response 5

Thank you very much for your suggestion. The changes have been implemented successfully.

Comment 6

Line 227-229: comparative fit index (CFI), the normalised fit index (NFI), the incremental fit index (IFI) 227 and the Tucker-Lewis index (TLI) obtained scores above 0.910. In addition, the root mean 228 square error of approximation analysis (RMSEA) obtained a value of 0.037. Line 248: (TC) (p≤0.05; r=0.204) Replace by: (TC; p≤.05; r=.204). Make the same change on the following lines

Response 6

Thank you very much for your suggestion. The changes have been implemented successfully.

Reviewer 3 Report

Nice paper and idea. I recommend acceptance

Author Response

REVIEWER 3

Comment 1

Nice paper and idea. I recommend acceptance

Response 1

Thank you very much for your comment. In this case the research has been carried out with the highest possible scientific rigour.